# FlexGen: Flexible Multi-View Generation from Text and Image Inputs

## Abstract

In this work, we introduce FlexGen, a flexible framework designed to generate controllable and consistent multi-view images, conditioned on a single-view image, or a text prompt, or both. FlexGen tackles the challenges of controllable multi-view synthesis through additional conditioning on 3D-aware text annotations. We utilize the strong reasoning capabilities of GPT-4V to generate 3D-aware text annotations. By analyzing four orthogonal views of an object arranged as tiled multi-view images, GPT-4V can produce text annotations that include 3D-aware information with spatial relationship. By integrating the control signal with proposed adaptive dual-control module, our model can generate multi-view images that correspond to the specified text. FlexGen supports multiple controllable capabilities, allowing users to modify text prompts to generate reasonable and corresponding unseen parts. Additionally, users can influence attributes such as appearance and material properties, including metallic and roughness. Extensive experiments demonstrate that our approach offers enhanced multiple controllability, marking a significant advancement over existing multi-view diffusion models. This work has substantial implications for fields requiring rapid and flexible 3D content creation, including game development, animation, and virtual reality.

## 1 Introduction

Recent progress in generative models (Song et al., 2020b; Ho et al., 2020) has significantly advanced 2D content creation, thanks to the rapid increase in 2D data volumes. However, 3D content creation remains challenging due to the limited accessibility of 3D assets, which are essential for diverse downstream applications, including game modeling (Gregory, 2018; Lewis & Jacobson, 2002), computer animation (Parent, 2012; Lasseter, 1987), and virtual reality (Schuemie et al., 2001). Previous 3D generation methods primarily concentrate on optimization-based techniques using multi-view posed images (Wang et al., 2021; Yariv et al., 2021; Ge et al., 2023; Verbin et al., 2022), or employ SDS-based distillation approaches derived from 2D generative models (Lin et al., 2023; Poole et al., 2022; Liang et al., 2023). Although effective, these methods often demand significant optimization time, limiting their practicality in real-world applications.

Multi-view diffusion models (Liu et al., 2023; Long et al., 2024; Shi et al., 2023a;b; Wang & Shi, 2023) have demonstrated the potential of pre-trained 2D generative models for 3D content creation through the synthesis of consistent multi-view images. Despite their promising performance, the controllable generation of multi-view images remains under-explored. Most existing multi-view diffusion models typically rely on a single-view image, which lacks 3D-aware controllable guidance and proves insufficient for robust multi-view image generation. For example, the generation of unseen regions continues to pose a significant challenge, often simply replicating information from the input view to unseen areas.

Several studies focus on conditional 3D generation. For instance, Coin3D (Dong et al., 2024) utilizes basic shapes as 3D-aware guidance, whereas Clay (Zhang et al., 2024) leverages sparse point clouds and 3D bounding boxes. However, these guidance methods are not user-friendly. Other research (Xu et al., 2024b;a) employs sparse multi-view images as 3D-aware guidance for 3D generation. These methods effectively supplement most unseen regions due to the incorporation of additional views, thereby achieving promising results. Nevertheless, acquiring sparse multi-view images is not always feasible in practice. These approaches typically depend on a pre-trained multi-view diffusion model

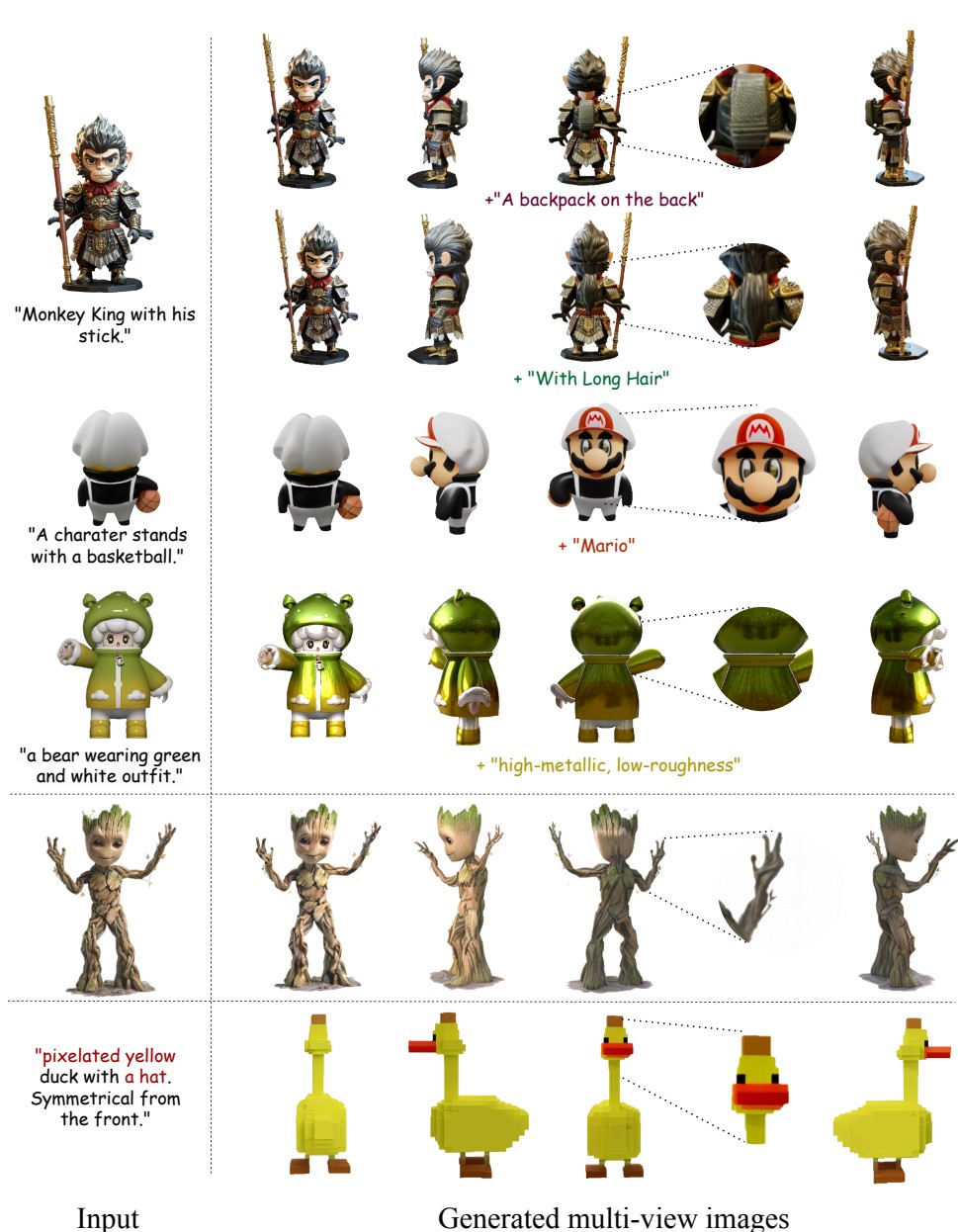

Figure 1: FlexGen is a flexible framework designed to generate high-quality, consistent multi-view images conditioned on a single-view image, or a text prompt, or both. Our method allows editing of unseen regions and modification of material properties through user-defined text.

for this purpose, yet integrating 3D-aware controllable guidance into multi-view diffusion models remains a significant challenge. Inspired by controllable content creation in 2D domain with text prompts, similar design can be incorporated as extra condition for providing 3D-aware guidance. Text encompasses adequate relationship information, which has been proven in the 2D generative model (Shen et al., 2024). Previous SDS-based distillation methods (Liang et al., 2023; Lin et al., 2023; Poole et al., 2022; Wang et al., 2024) have successfully employed text prompts to guide 3D asset generation, yielding significant results and validating this approach.

However, designing 3D-aware text annotations for 3D assets is not trivial. Most prior works (Li et al., 2023c; Achiam et al., 2023) focus on 2D image captioning, which lacks sufficient 3D-aware information. To overcome this limitation, Cap3D (Luo et al., 2023) leverages the reasoning capabilities of BLIP-2 (Li et al., 2023c) to generate descriptions for each rendered view of a 3D asset. These descriptions are then aggregated by GPT-4 (Achiam et al., 2023) to form a comprehensive caption. However, this approach often results in a high-level summary that misses detailed local captions. This limitation arises from two main factors: firstly, BLIP-2 primarily produces global descriptions, and secondly, individual view provides limited information about the object, often resulting in redundant or incomplete single-view annotations. Instant3D (Li et al., 2023a) is the most similar work to ours, utilizing text prompts from Cap3D to generate multi-view images. However, as discussed before, text annotations from Cap3D lack 3D-aware information and Instant3D only supports text-to-3D task, which is not flexible enough.

Alternatively, we propose leveraging the powerful recognition capabilities of GPT-4V(ision) to perform 3D-aware global-to-local captioning. By analyzing four orthogonal rendered views from a object, GPT-4V is capable of generating detailed descriptions that capture both global context and local features with 3D-aware information. This approach ensures the resulting text annotations are enriched with 3D-aware details with spatial relationship, providing a more comprehensive and perceptually accurate understanding. We show a comparison with Cap3D in Figure 2.

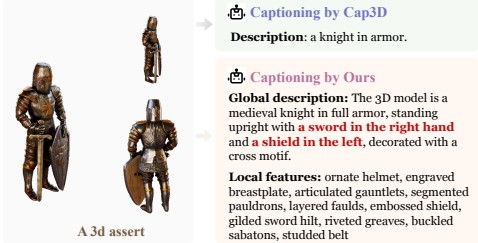

Figure 2: Comparison of the caption between Cap3D and Ours.

With 3D-aware text annotations, we can enhance the generative capabilities of existing multi-view diffusion models, enabling controllable generation guided by text prompts. By modifying the text prompt, our model is capable of generating diverse and consistent multi-view images that accurately correspond to the given textual descriptions. To this end, we propose an adaptive dual-control module that adaptively integrates image and text modalities through reference attention, enabling precise joint control over multi-view image generation by leveraging both visual input and detailed text prompts. In this work, we demonstrate three kinds of controllability of our model: firstly, our model can supplement unseen parts, achieving controllable unseen part generation; secondly, we introduce material controllability by adjusting the text prompt to modify the metallic and roughness properties when rendering multi-view images. For instance, appending the prompt with "high metallic" and "low roughness" can reflect these material characteristics accurately; finally, our model enables part-level control over the texture of the generated unseen parts, as we incorporate detailed texture information during the text annotation phase.

To summarize, our contributions are listed as follows.

- We propose FlexGen, a method for flexible generation of multi-view images, guided by both image and text inputs. This approach offers robust controllability and ensures that the generated images accurately correspond to the given textual descriptions.

- We annotate 3D-aware text guidance using GPT-4V and generate material prompts consistent with the materials used during rendering, achieving controllable generation of textures, materials, and unseen parts.

- Extensive experiments validate the effectiveness of our proposed methods, demonstrating the ability to controllable generation of multi-view images.

## 2 RELATED WORK

### 2.1 DIFFUSION MODELS FOR MULTI-VIEW SYNTHESIS

Recent research has extensively explored the generation of multi-view images using diffusion models to achieve efficient and 3D-consistent results. These efforts include both text-based methods, such as MVDiffusion (Deng et al., 2023), MVDream (Shi et al., 2023b) and ImageDream (Wang & Shi, 2023), and image-based methods like SyncDreamer (Liu et al., 2023), Wonder3D (Long et al.,

2024) and Zero123++ (Shi et al., 2023a). MVDiffusion, for instance, leverages text conditioning to simultaneously generate all images with a global transformer, facilitating cross-view interactions. Similarly, MVDream incorporates a self-attention layer to capture cross-view dependencies, ensuring consistency across different views. For image-based approaches, SyncDreamer constructs a volume feature from the multi-view latent representation to produce consistent multi-view color images. Wonder3D enhances the quality of 3D results by explicitly encoding geometric information and employing cross-domain diffusion. Several methods (Tang et al., 2024; Zhang et al.; Zheng et al., 2024; Xu et al.) adopt these approaches to first generate multi-view images of an object and then reconstruct the 3D shape from these views using sparse reconstruction techniques. Despite achieving reasonable results, these methods still face limitations in controllable generation.

## 2.2 CONTROLLABLE GENERATIVE MODELS

In recent years, adding conditional control to generative models has garnered increasing attention for enabling controllable generation. These efforts span both 2D and 3D domains. For instance, several 2D methods (Hess, 2013; Brooks et al., 2023; Gafni et al., 2022; Kim et al., 2022; Parmar et al., 2023) focus on text-guided control by adjusting prompts or manipulating CLIP features. Additionally, ControlNet (Zhang et al., 2023) allows for a series of 2D image hints for control tasks through a parallel model architecture. However, similar controllable capabilities in 3D generation (Bhat et al., 2024; Cohen-Bar et al., 2023; Pandey et al., 2024) remain largely inapplicable. Coin3D (Dong et al., 2024) introduces a framework for generating 3D assets guided by basic shapes and text. Nonetheless, basic 3D shapes are not user-friendly for general users, whereas text serves as a more intuitive and accessible conditional input. In this work, we integrate text prompts into the multi-view diffusion model for controllable multi-view image generation, enabling the model to generate controllable unseen regions when a single-view image is simultaneously provided.

## 3 METHOD

FlexGen is a flexible multi-view generation framework that supports conditioning based on text, single-view images, or a combination of both. By incorporating 3D-aware text annotations derived from GPT-4V, our method effectively achieve controllable multi-view images generation, including reasonable unseen part generation, texture controllable generation and materials editing. We begin with a succinct problem formulation in Section 3.1. Then, we introduce how to annotate 3D-aware caption in Section 3.2. After that, adaptive dual-control module is introduced to add textual condition into the framework in Section 3.3. Finally, we introduce training and inference in Sectin 3.4. An overview framework of FlexGen is shown in Figure 3.

## 3.1 PROBLEM FORMULATION

Given a single-view image $I$ or a user-defined prompt $T$ that describes an object or both, our goal is to develop a generative model $G$ that produces a tiled image $I_{out}$. This image consists of a $2 \times 2$ layout, representing 4 views of an object - the front, left, back, and right - with each view at a resolution of $512 \times 512$. The multi-view images are aligned with both the single-view image and the prompt, ensuring consistency among them, as illustrated in Figure 1. Inspired by (Li et al., 2023b), our generative model $G$ is based on a large pre-trained text-to-image diffusion model. The design of the $2 \times 2$ image grid aligns better with the original data format used by the 2D diffusion model, facilitating the utilization of more prior knowledge. Regardless of the focal length and pose of the input image $I$, we consistently generate orthographic images with a fixed focal length and an elevation angle of $5°$. For example, when processing an input image captured at an elevation angle $\alpha$ and azimuth angle $\beta$, FlexGen generates multi-view images at azimuth angles $\{\beta, \beta + 90°, \beta - 90°, \beta + 180°\}$, all with a fixed elevation of $5°$.

## 3.2 3D-AWARE CAPTION ANNOTATION

Cap3D (Luo et al., 2023) utilized BLIP (Li et al., 2023c) to annotate each single rendered view and then leverage GPT4 for holistic description. However, such method leads to text annotation lacking 3D-aware information. Alternatively, we construct a dataset consists of paired multi-view images and 3D-aware global-local text annotation, as shown in the Figure 4. Our dataset is based

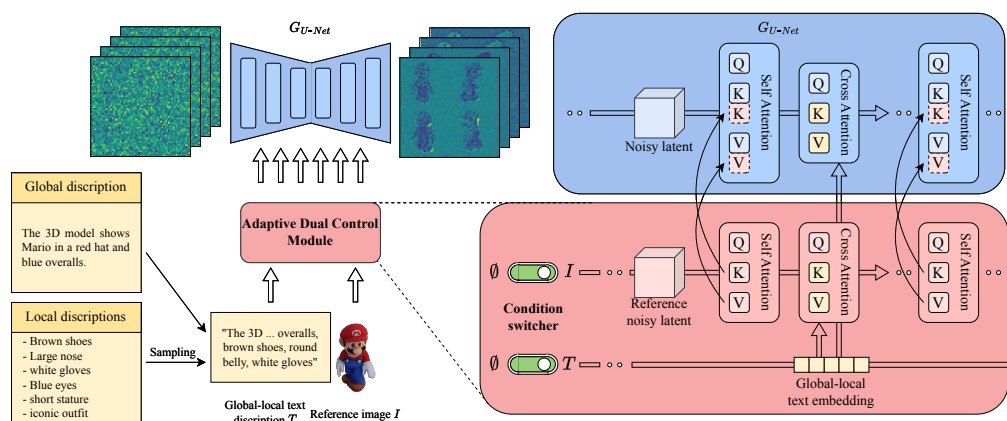

Figure 3: **Overview of the framework.** FlexGen is a flexible framework to generate controllable and consistent multi-view images, conditioned on a single-view image, or a text prompt, or both. The system incorporates a 3D-aware annotation method using GPT-4V and an adaptive dual-control module that integrates both a reference input image and text prompts for precise joint control. The condition switcher enhances flexibility, enabling the model to generate multi-view images based on image input, text input, or a combination of both modalities.

on Objaverse (Deitke et al., 2023), which provides basic shape and textual for each object, lacking high-quality textual description for each 3D asset. Therefore, we utilized an advanced large-scale multimodal model, GPT-4V, to generate high-quality textual descriptions for each 3D asset. Given these orthogonal views, GPT-4V not only summarizes a comprehensive global description of the object but also captures the intricate 3D relationships between its components. Specifically, the dataset construction process consists of three steps: rendering, captioning, and merging. (1) In the rendering stage, each 3D asset is rendered into four orthogonal multi-view images with a resolution of 512×512, which are tiled into a single image $I_{out}$ in a $2 \times 2$ layout. (2) The captioning step is performed using GPT-4V with tiled image, which generates 3D-aware global and local captions. (3) In the final step, the global and local descriptions are merged to form the "global-local text description" of the 3D asset. During training, we randomly select a portion of the local descriptions to simulate user behavior.

Moreover, we incorporate material descriptions, such as metallic and roughness attributes, into the text annotations to enable material-controllable generation. We propose adding material descriptions that correspond to the materials used during rendering by blender Hess (2013). For example, if multi-view images are rendered with high metallic and low roughness, we enhance the prompt with "high metallic" and "low roughness" to ensure the descriptions match the visual data. More details on the material rendering can be found in the Appendix.

## 3.3 ADAPTIVE DUAL-CONTROL MODULE

Previous approaches, such as those by Li et al. (2023a) and Shi et al. (2023a), typically focus on single-modality inputs, conditioning solely on either a text prompt or a single-view image, without enabling joint control over both for generating multi-view images. To overcome this limitation, we propose an adaptive dual-control module that allows for simultaneous conditioning on both image and text inputs, enabling more precise and flexible multi-view image generation.

Our method builds upon the reference attention mechanism (Zhang, 2023), which we extend to integrate both the reference image and text prompt effectively. This enhanced integration facilitates robust interaction between the two modalities, enabling our model to generate multi-view images that maintain (1) high fidelity to the input image and (2) coherence with both the global and fine-grained descriptions specified in the text

Reference attention involves running the denoising UNet model on an additional reference image, appending the self-attention key and value matrices from the reference image to the corresponding

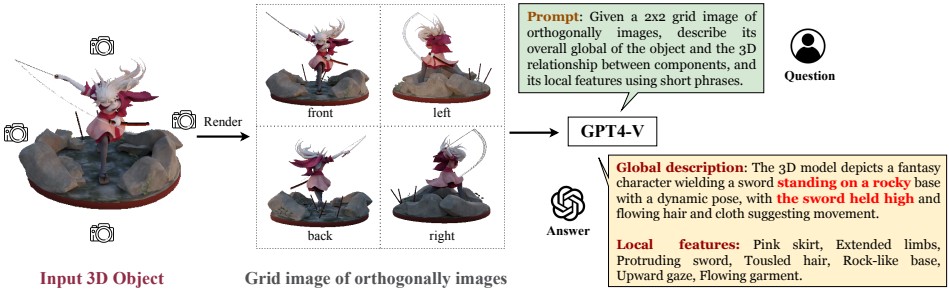

Figure 4: **3D-aware caption generation pipeline.** A 3D object is rendered into four orthogonal multi-view images (front, left, back, right) in a $2 \times 2$ grid layout. Using GPT-4V, the agent generates both global and local descriptions. The global description captures the overall attributes of the object and the 3D spatial relationships between its components, while local features detail specific aspects such as color, posture, and texture, thereby enriching the dataset with rich semantic annotations.

attention layers during model denoising. We introduce a slight modification to this approach. In addition to the reference image, we inject the prompt information. Specifically, the user-defined prompt is processed through the CLIP encoder to obtain per-token CLIP text embeddings $E$ with a shape of $L \times D$, where $L$ represents the token length and $D$ the embedding dimension. This prompt includes both global descriptions and local features. Using the cross-attention mechanism, we facilitate sufficient information interaction between the image and prompt, enabling more precise joint control. Once this interaction is complete, we append the self-attention key and value matrices from our adaptive dual-control module to the corresponding attention layers during model denoising.

## 3.4 TRAINING AND INFERENCE

Building on the adaptive dual-control module, our framework accommodates both prompt and image conditions to guide the generation of multi-view images. To increase flexibility, we introduced a condition switcher during training that supports both single-mode and dual-mode conditions, allowing for seamless transitions between different input scenarios. With a configurable probability, inputs can be left empty: when the text prompt is absent, it defaults to an empty string to ensure uninterrupted processing by the model. Similarly, in the absence of an image input, we substitute it with a black image.

During inference, this design facilitates flexible and controllable multi-view generation. When both modalities are available, the image and prompt collaborate to provide complementary information, enriching the generated output. If only the image is supplied, the model functions in an image to multi-view mode, generating multiple views based solely on the visual input. Conversely, when only the text is available, the model operates as a text to multi-view generator, producing views that align with the textual description.

## 4 EXPERIMENTS

### 4.1 EVALUATION SETTINGS

**Training Datasets.** Given the inconsistent quality of the original Objaverse dataset (Deitke et al., 2023), we initially excluded objects lacking texture maps and those with low polygon counts. We subsequently curated a collection of 147k high-quality objects to form the final training set. For rendering these objects, we employed Blender, setting the camera distance at 4.5 units and the field of view (FOV) to 30 degrees. We generated 24 ground-truth images for the target view set, maintaining a fixed elevation of 5 degrees while uniformly distributing the azimuth angles across the range [0, 360]. The input views were randomly sampled with elevation angles between -30 and 30 degrees and azimuth angles evenly distributed across [0, 360]. All images were rendered at a resolution of $512 \times 512$. For the purpose of 3D-aware caption annotation, we utilized four orthogonal images from the ground-truth set as inputs for GPT-4V.

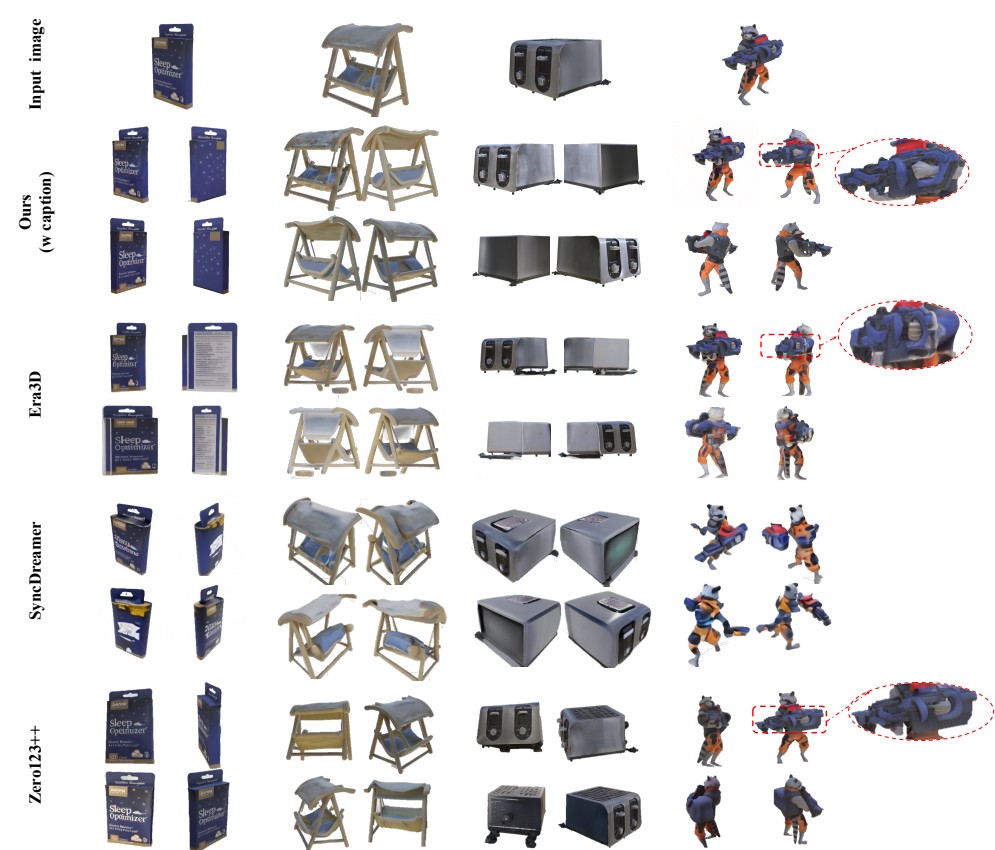

Figure 5: **Qualitative comparison of novel view synthesis.** Our method achieves superior generation quality by integrating both text and image guidance. The text prompts used in our approach are generated by GPT-4V based on the input view. Some details are enlarged in the red circle

**Training Details.** We utilized Stable Diffusion 2.1 as our base model, training it on eight NVIDIA A800 80GB GPUs over a period of 10 days, completing 180,000 iterations with a batch size of 32. The Adam optimizer was employed with a learning rate of 1e5. During training, we configured the probabilities of using both the prompt and image, only the image, and only the text at 0.3 each, while the probability for both modalities being absent was set at 0.1. Sampling involved 75 steps using the DDIM methodology (Song et al., 2020a).

Table 1: The quantitative comparison in novel view synthesis and sparse-view reconstruction. We report PSNR, LPIPS, CD and FS on the GSO dataset.

| Method | PSNR↑ | LPIPS↓ | CD↓ | FS@0.1↑ |
|---|---|---|---|---|
| Ours | 22.31 | 0.12 | 0.076 | 0.928 |
| Ours(w/o caption) | 21.12 | 0.14 | 0.078 | 0.921 |
| Zero123++ | 18.83 | 0.16 | 0.087 | 0.910 |
| Era3D | 18.52 | 0.19 | 0.245 | 0.713 |
| SyncDreamer | 17.66 | 0.21 | 0.126 | 0.833 |

Table 2: The quantitative comparison with MVDream in text to multi-view synthesis. "Ground truth" refers to the multiple views used to generate text by GPT-4v. Our method outperform MVDream by a large margin.

| Method | FID ↓ | IS↑ | CLIP↑ |
|---|---|---|---|
| Ours | 35.56 | 13.41±0.87 | 0.83 |
| Ground truth | N/A | 13.81±1.40 | 0.89 |
| MVDream | 44.42 | 12.98±1.22 | 0.79 |

**Evaluation Metrics.** We evaluated our method across three distinct tasks: text to multi-view image generation, novel view synthesis (NVS), and sparse-view 3D reconstruction using the generated images as input to a reconstruction module. For text to multi-view generation, we employed the Frechet Inception Distance (FID) (Heusel et al., 2017) and Inception Score (IS) (Salimans et al., 2016) to assess image quality, and CLIP score (Radford et al., 2021) to evaluate alignment between the generated images and the textual descriptions. For NVS, we utilized LPIPS (Zhang et al., 2018) to compare perceptual similarity between the generated novel-view images and the ground truth. The

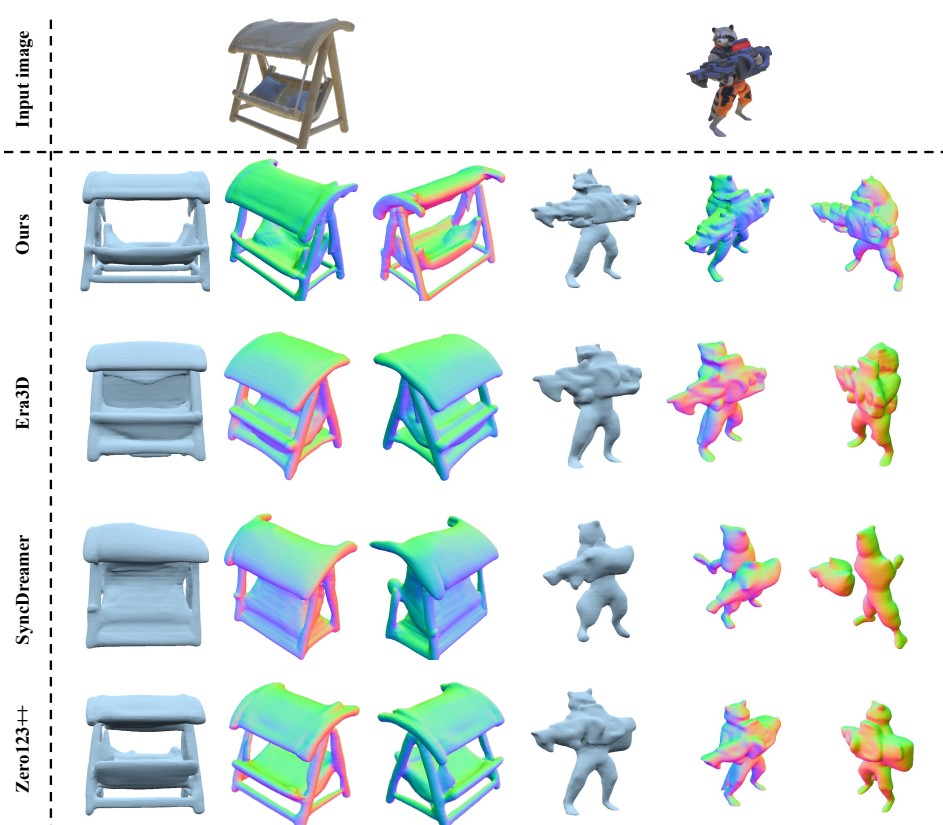

Figure 6: Qualitative comparison with Era3D, SyncDreamer and Zero123++ on sparse-view 3D reconstruction. Our method generates consistent multi-view images, achieving better results.

quality of 3D reconstruction was assessed using Chamfer Distance (CD) and volumetric Intersection over Union (IoU) between the reconstructed meshes and the ground truth ones.

**Evaluation Datasets** We employed the Google Scanned Object (GSO) dataset (Downs et al., 2022) for evaluation purposes. For the text to multi-view generation task, we randomly selected 300 samples from the GSO dataset. For each sample, we rendered a set of four orthogonal views and employed our captioning method to generate 300 text prompts that accurately describe the objects. These same 300 samples were also used for evaluation in the novel view synthesis (NVS) and sparse-view 3D reconstruction tasks.

## 4.2 COMPARISON WITH STATE-OF-THE-ART METHODS

**Novel view synthesis and sparse-view 3D reconstruction** First, we quantitatively compare our method with other single-image to multi-view approaches, including Zero123++ (Shi et al., 2023a), Era3D(Li et al., 2024), and SyncDreamer (Liu et al., 2023), as shown in Table 1. Our method outperforms the others across several key metrics, such as PSNR and LPIPS, demonstrating that the joint control of text prompts and images allows for more consistent and realistic multi-view generation, particularly in unseen areas. Qualitative results are presented in Figure 5. To further verify the consistency of multi-view generation in three dimensions, we reconstructed the multi-view images generated by all methods using the open-source reconstruction method InstantMesh (Xu et al., 2024b) for a fair comparison. We report Chamfer Distance (CD) and FS metrics in Table 1, which show that our multi-view images lead to more accurate geometry reconstructions. Qualitative comparison of 3D reconstruction results are shown in Figure 6.

**Text to multi-view.** For text to multi-view evaluation, we conducted both qualitative and quantitative comparisons with the only existing open-source model, MVDream (Shi et al., 2023b). Table 2

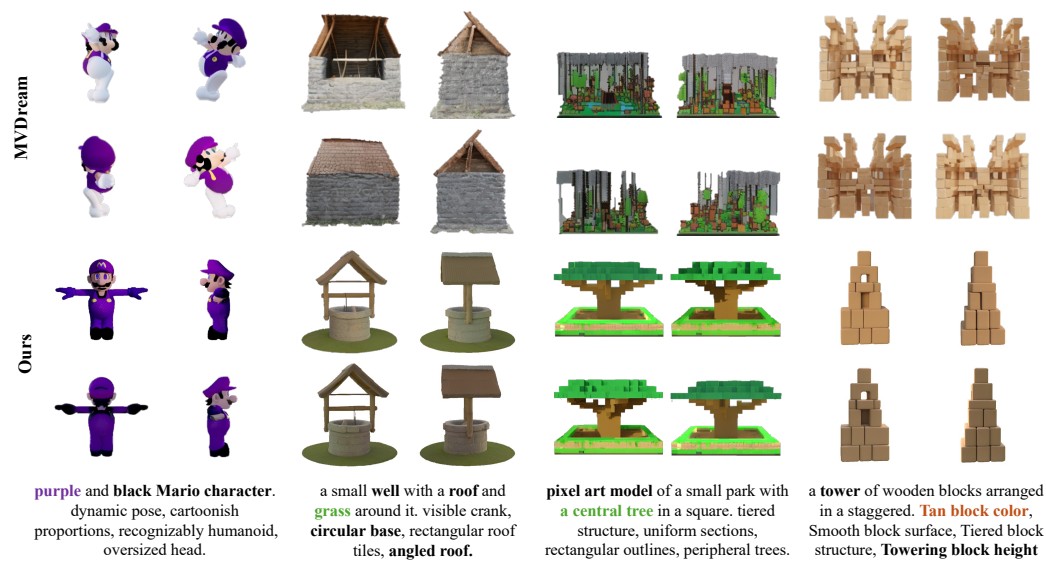

Figure 7: Qualitative comparison of text to multi-view. Our method significantly outperforms MV-Dream, achieving better generation quality that is consistent with the text.

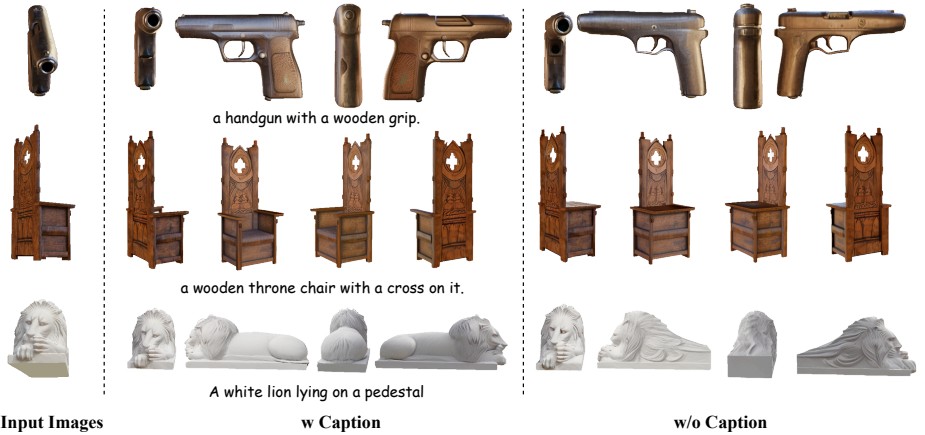

Figure 8: Ablation study to demonstrate that caption is capable of supplement unseen part, achieving significant better and reasonable results.

presents quantitative results, highlighting differences in generation quality and text-image consistency. Our model achieved Inception Score (IS) and CLIP score metrics that were comparable to those of the validation set, demonstrating strong image quality and text-image alignment. Furthermore, our model consistently outperformed MVDream across all evaluated metrics. Figure 7 showcases qualitative examples from the validation set, our model, and MVDream, illustrating that our model produces higher-quality, multi-view consistent images that more accurately match the text prompts. The images from our model exhibit superior fidelity and adherence to the input descriptions compared to those generated by MVDream, underscoring the effectiveness of our approach.

### 4.3  ABALTION STAUDY

We conducted an ablation study to evaluate the impact of key components in our approach, specifically focusing on the 3D-aware caption annotation and the Adaptive dual-control module.

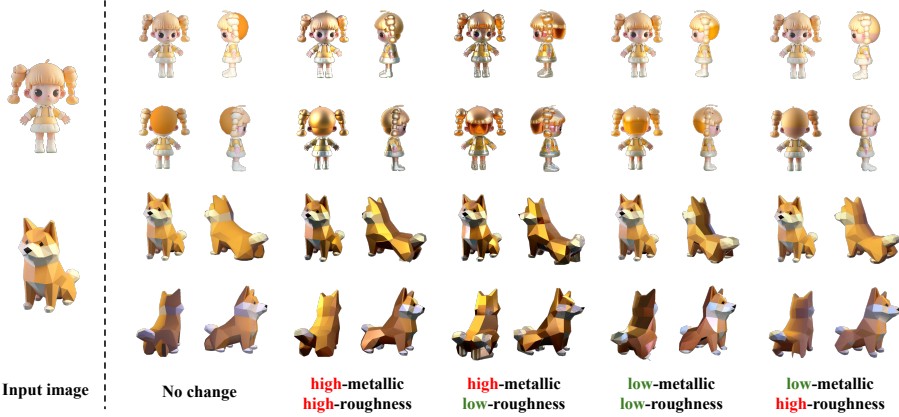

Figure 9: Material editing in generated multi-view images is facilitated by providing prompts such as "high-metallic" and "low-roughness".

**Adaptive Dual-Control Module.** The adaptive dual-control module represents a key innovation within our framework, enabling simultaneous control over both image and text inputs. Unlike previous models that typically focus on a single modality, our approach allows for enhanced flexibility and precision in generating outputs. This dual-input capability empowers users to guide the generation process using both image and textual prompts, significantly enhancing the model's ability to produce coherent and contextually appropriate multi-view images, as demonstrated in Figure 8. This integration of modalities marks a substantial advancement over existing controllable generative models, offering a more intuitive and effective means for users to influence the output. Additionally, we illustrate the module's capability to edit material properties in the generated multi-view images, as shown in Figure 9.

**The impact of 3D-Aware Captioning.** To assess the effectiveness of 3D-aware captioning, we trained our model using annotations from the Cap3D dataset and our own dataset separately, comparing the control capabilities afforded by image and text prompts. As illustrated in Figure 10, training with Cap3D data limits the ability for fine control, whereas our method, through its integration of 3D-aware information, enables more effective editing of text prompts.

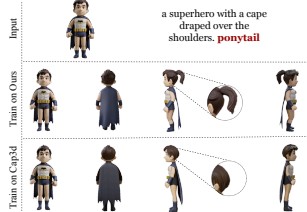

Figure 10: Ablation of 3D-aware Captioning .

## 5 CONCLUSION

In this work, we present FlexGen, a multi-view diffusion model designed to generate consistent and controllable multi-view images guided by a single image, text, or a combination of both. To achieve this, we harness the powerful recognition capabilities of GPT-4V to perform 3D-aware text annotations by reasoning over orthogonal views of an object, arranged as tiled multi-view images. Additionally, we introduce an adaptive dual-control module that enables text-based conditioning to be incorporated directly into the generation phase. By embedding spatial relationships, texture, and material descriptions into the text annotations, we achieve enhanced control over the output. Extensive experiments demonstrate the effectiveness of our proposed approach.

**Limitations and future works.** Although FlexGen introduces the innovative capability to jointly control both image and text inputs, our method occasionally encounters difficulties with complex user-defined instructions. This limitation likely stems from the constraints imposed by the availability of high-quality datasets. In the future, we plan to expand the dataset size and enhance the control capabilities to more effectively accommodate intricate instructions.

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

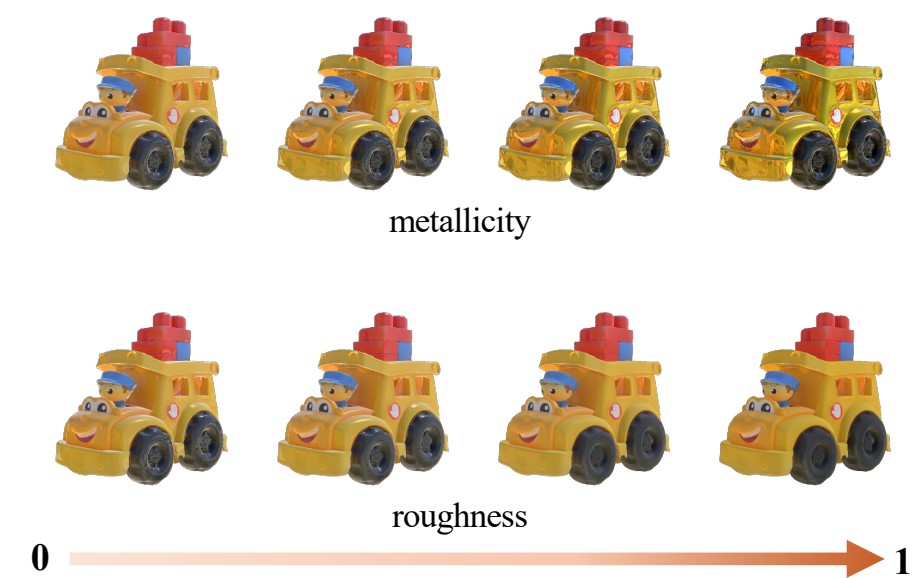

metallicity

roughness

**0** ————————————————————————→ **1**

Figure 11: Visualization of an object rendered with different values of metallicity and roughness.

## A APPENDIX

### A.1 DETAIL OF MATERIAL RENDERING

We integrate material descriptions, including attributes such as metallicity and roughness, into the text annotations to enable material-controllable generation. These descriptions are designed to correspond with the materials utilized during rendering in Blender. With Blender, we can freely adjust the values of metallicity and roughness, allowing us to render corresponding images, as shown in the figure. The values for metallicity and roughness range from 0 to 1. Specifically, when the values are below 0.3, the corresponding prompt is "low" and when they exceed 0.6, the corresponding prompt is "high".

### A.2 MORE VISUALIZATION RESULTS

We show more visualization of our model in Figure **??**.

### A.3 FAILURE CASES

Figure 13 illustrates two failure cases, where the input text is unable to serve as an editor for multiple views. This limitation arises due to the absence of relevant text descriptions in the training data.

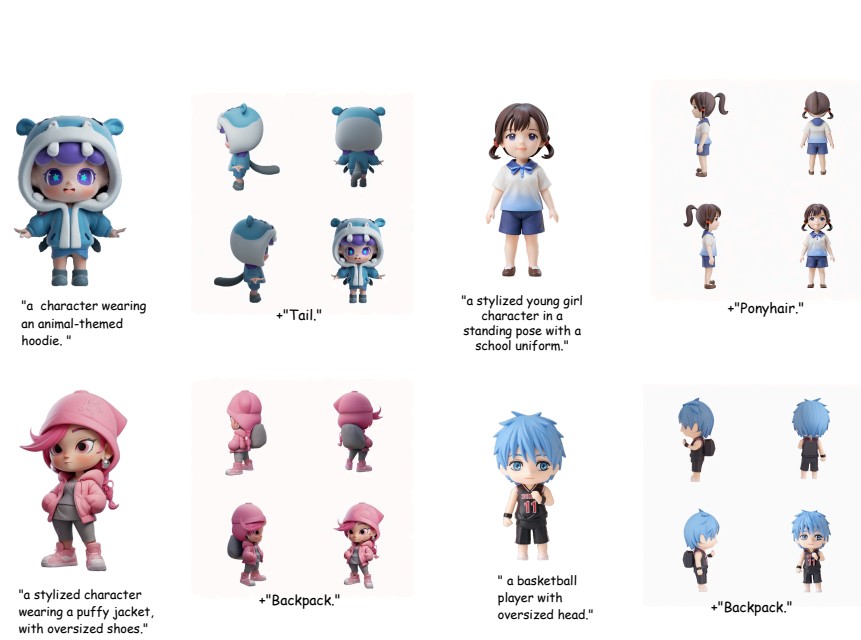

Figure 12: illustration of some success cases

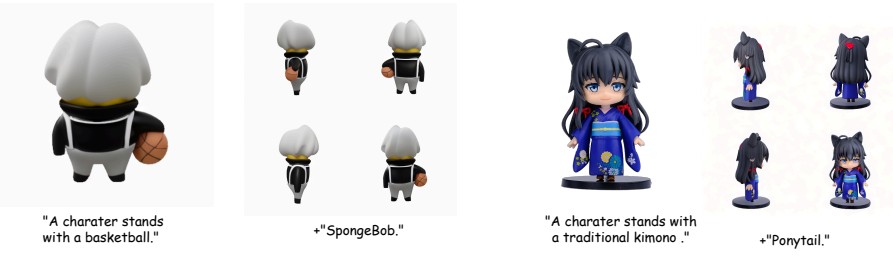

Figure 13: illustration of some failure cases

