# OpenReview forum: "FlexGen: Flexible Multi-View Generation from Text and Image Inputs"
_ICLR.cc/2025/Conference — ICLR 2025 Conference Withdrawn Submission_

### Official Review · Reviewer_GERz · 2024-10-28

**Soundness:** 3
**Presentation:** 2
**Contribution:** 1
**Rating:** 3
**Confidence:** 4

**Summary:**

This work annotates Objaverse with GPT4v and trains a mulit-view generation from both image and text inputs.

**Strengths:**

- They annotate Objaverse by GPT4v which will be a good addition to the community if the authors would like to open source.

**Weaknesses:**

- Limited technical novelty. ImageDream (Wang and Shi, 2023) trained a multi-view image generation from both image and text and showed similar capability. Note their method can also be used to add new unseen details at the back, check their opensourced code here: https://github.com/bytedance/ImageDream. I find the using both image and text and the shared attention mechanisms are close in two works.  My suggestion: (1) compare to them technically; (2) show ablation study why your design is better than theirs.

-  Lacks mathematical formulation for Adaptive Dual-Control Module. What is the formulation of condition switcher in Fig. 3?  From Sec. 3.4, is the switcher a simple dropout training and zero inference if missing? Consider providing a formal mathematical description of the Adaptive Dual-Control Module, including the condition switcher. This would enhance the technical depth of the paper and allow for better reproducibility.

- Missing ablation study.  Could you include ablation studies that address: (1) The impact of using the curated vs. regular Objaverse dataset, (2) The importance of the GPT4v caption in the model's performance, (3) The effect of injecting into both self-attention and cross-attentions? and (4) other designs that are crucial / novel to your work. These studies would help readers understand the relative importance of each component in your method."

**Questions:**

- Will you release your annotation?

- Can you elaborate on the key architectural differences between your model and other multiview diffusion models like Zero123++? From the numbers and visuals, seems the FlexGen outperforms the previous art by a large margin. But from the technical side, there seems to be not clear significant difference with previous work. What specific components or techniques in your approach contribute most significantly to the improvements you observe?

---

### Official Review · Reviewer_X4qv · 2024-10-29

**Soundness:** 3
**Presentation:** 3
**Contribution:** 3
**Rating:** 6
**Confidence:** 4

**Summary:**

This paper introduces FlexGen, a novel framework for generating consistent and controllable 4 multi-view images from single-view images, text prompts, or both. The key contributions are: (1) A captioning pipeline that utilizes GPT-4V to generate 3D-aware text annotations from rendered orthogonal views. (2) A new framework that integrates image and text modalities for fine-grained control over the generation process. The results are solid, showing clear performance gains over recent baselines.

**Strengths:**

1: The proposed framework is flexible, supporting generation from single-view images, text prompts, or both. This allows for versatile applications and user interactions. The adaptive dual-control module enables fine-grained control over various aspects of the generated multi-view images, including unseen parts, material properties, and textures, showcasing impressive controllability compared to existing methods.

2: The paper presents extensive experiments on the Objaverse and GSO datasets, demonstrating superior performance in novel view synthesis, sparse-view 3D reconstruction, and text-to-multi-view generation compared to state-of-the-art methods.

3: The writing is very clear and easy to follow. However, the appendix could be slightly refined, as it seems to have been rushed before the deadline.

**Weaknesses:**

1: While GPT-4V enables rich 3D-aware annotations, generating these can be computationally expensive and relies on a proprietary model. Exploring open-source MLLMs for captioning could be valuable, potentially increasing accessibility and reducing dependence on closed-source solutions. The paper could benefit from discussing the trade-offs between annotation quality and computational cost when using different models.

2: The paper mentions occasional difficulties with complex user-defined instructions. Further investigation is needed to understand the limitations of the current approach and improve its robustness in handling complex scenarios. Including visual examples would be beneficial.

3: A key limitation is the fixed 4-view output. While sufficient for some tasks, it falls short compared to video-diffusion models like Emu-Video used in im-3d and vfusion3d (16 views) or SV3D (20 views). Additionally, FlexGen cannot synthesize novel views from arbitrary angles, a capability demonstrated by SV3D and Cat3D. This restricts its use in applications requiring more comprehensive 3D understanding or flexible viewpoint control.

**Questions:**

1: The paper primarily focuses on generating a fixed set of views. Did the authors considered enabling novel view synthesis with arbitrary viewing angles, similar to methods like SV3D or Cat3D? If so, how to envision adapting FlexGen to achieve this?

I find the core idea of this paper interesting and the presented results are solid. I am currently tending towards a borderline accept prior to the rebuttal. I believe the paper has the potential to make a significant contribution to the field, but I would like to see the authors address the raised weaknesses and questions in their rebuttal to solidify my decision.

---

### Official Review · Reviewer_9rpW · 2024-11-03

**Soundness:** 2
**Presentation:** 2
**Contribution:** 2
**Rating:** 3
**Confidence:** 5

**Summary:**

This paper proposes a method for generating multi-view images conditioned on both image and text prompts. Building on the reference attention mechanism from prior work, it incorporates additional text conditioning to enable controllable generation through text prompts. To enhance text captions, the authors use GPT-4V to annotate 3D assets and render objects with two different material properties, allowing for varied material appearances. Quantitative experiments demonstrate improved performance over several baseline methods in view synthesis, 3D reconstruction, and text-to-multi-view tasks.

**Strengths:**

- The use of GPT-4V for multi-view annotation is effective and shows promising results.
- The paper explores an interesting approach by incorporating material properties into multi-view synthesis within generative models.

**Weaknesses:**

- The primary contributions rely on integrating existing techniques (GPT-4V for captioning and a previously established reference-guided mechanism), rather than proposing fundamentally new methodologies.
- While the detailed captioning using GPT-4V is beneficial, the approach does not introduce a novel annotation strategy beyond leveraging GPT’s generative capacity.
- The core of the proposed approach relies heavily on previously established methods, specifically the reference view guidance. The main component, known as the "key-value (k, v) appending mechanism," which enables reference view guidance, was first introduced in prior work by Zhang et al. [1]. This paper primarily extends the mechanism by adding additional text prompts for control, but even this extension is not entirely novel; the concept of using text prompts alongside multi-view guidance has been previously explored in works such as Direct2.5 [2] and MVControl [3].
- The paper lacks quantitative evaluation to assess the effectiveness of material properties (e.g., metallic, roughness). Additionally, the approach does not appear scalable, as supporting a new material requires generating an entirely new set of images in Blender. It is also challenging to predict how well material conditioning via text prompts would perform for a broader range of material combinations.

[1] Lyumin Zhang. Reference-only control. In Reference-only control, pp. https://github.com/Mikubill/sd–webui–controlnet/discussions/1236. github, 2023.

[2] Lu, Y., Zhang, J., Li, S., Fang, T., McKinnon, D., Tsin, Y., Quan, L., Cao, X. and Yao, Y., 2024. Direct2. 5: Diverse text-to-3d generation via multi-view 2.5 d diffusion. In Proceedings of the IEEE/CVF Conference on Computer Vision and Pattern Recognition (pp. 8744-8753).

[3] Li, Z., Chen, Y., Zhao, L. and Liu, P., 2023. Mvcontrol: Adding conditional control to multi-view diffusion for controllable text-to-3d generation. arXiv preprint arXiv:2311.14494.

**Questions:**

- Given that much of the proposed approach relies on established techniques, could the authors clarify what specific aspects of the methodology are novel? In what ways does the integration of text prompts and reference guidance go beyond previous work.
- While GPT-4V is used for enhanced captioning, did you consider alternative or custom annotation strategies to achieve richer or more context-specific annotations for 3D assets? If so, why were they not pursued, and if not, how might they enhance your model’s performance?
- Currently, the approach requires generating new images in Blender for each material property. Have you considered any strategies to make the model more scalable in terms of material variations, possibly by automating or simulating material properties in the model itself?
- How does your model handle situations where the text and image prompts may conflict or suggest different visual details? Have you tested such cases, and if so, what did you observe about the model’s ability to reconcile or prioritize these inputs?

---

### Official Review · Reviewer_hSWB · 2024-11-05

**Soundness:** 2
**Presentation:** 2
**Contribution:** 2
**Rating:** 5
**Confidence:** 4

**Summary:**

The paper presents FlexGen, a framework designed for multi-view image synthesis using single-view images, text prompts, or both. The core methodology leverages GPT-4V to generate 3D-aware text annotations, aiming to achieve more controllable and consistent multi-view image generation.

**Strengths:**

+ FlexGen’s use of GPT-4V for 3D-aware captioning and the adaptive dual-control module offers flexibility in image synthesis, enabling detailed control over multi-view consistency and visual attributes.

**Weaknesses:**

- The main contribution appears to be the use of GPT-4V for generating detailed captions in multi-view synthesis. This application of existing technology lacks significant innovation and may not constitute a substantial advancement in multi-view generation.
- Qualitative results in Figures 5 and 6 do not clearly demonstrate a marked advantage of FlexGen over existing methods.
- Appendix Section A.2 lacks the corresponding figures and analysis that could further clarify the model’s performance and visual outputs.

**Questions:**

1. Does "Zero123++" refer to Zero123-XL? If not, could you clarify why Zero123-XL was not included for comparison?
2. How does this paper address the Janus problem in multi-view image synthesis?
3. Given FlexGen’s reliance on GPT-4V for 3D-aware captions, how does this approach overcome the limitations of generating multi-view consistency solely from text inputs?

**Details Of Ethics Concerns:**

No ethical issue.

---

### Note · Authors · 2024-11-15

I have read and agree with the venue's withdrawal policy on behalf of myself and my co-authors.